# DarKnight: A Data Privacy Scheme for Training and Inference of Deep Neural Networks

## Abstract

Protecting the privacy of input data is of growing importance as machine learning methods reach new application domains. In this paper, we provide a unified training and inference framework for large DNNs while protecting input privacy and computation integrity. Our approach called DarKnight uses a novel data blinding strategy using matrix masking to create input obfuscation within a trusted execution environment (TEE). Our rigorous mathematical proof demonstrates that our blinding process provides information-theoretic privacy guarantee by bounding information leakage. The obfuscated data can then be offloaded to any GPU for accelerating linear operations on blinded data. The results from linear operations on blinded data are decoded before performing non-linear operations within the TEE. This cooperative execution allows DarKnight to exploit the computational power of GPUs to perform linear operations while exploiting TEEs to protect input privacy. We implement DarKnight on an Intel SGX TEE augmented with a GPU to evaluate its performance.

## 1 Introduction

The need for protecting input privacy in Deep learning is growing rapidly in many areas such as health care (Esteva et al., 2019), autonomous vehicles (Zhu et al., 2014), finance (Heaton et al., 2017), communication technologies (Foerster et al., 2016) etc. Many of the data holders are, however, not machine learning experts. Hence, the need for machine learning as a service (MLaaS) has emerged. Microsoft Azure ML (Microsoft, 2020), Google AI platform (Google, 2020), Amazon ML (Amazon, 2020) are some examples. These services provide computing infrastructure and ML runtime to enable data holders to quickly set up their models and train. While these platforms accelerate the ML setup process, they exacerbate the user's concern regarding the data privacy.

In this paper, we propose DarKnight, a unified inference and training framework that protects data privacy with rigorous bounds on information leakage. DarKnight takes a unique hybrid-execution approach where it uses trusted execution environments(TEE) to blind input data using matrix masking techniques, and then uses GPUs to accelerate DNN's linear computations on the blinded data. Training or inference solely within TEEs can provide data privacy, by blocking access to TEE memory for all intruders, including root users, using hardware encryption and isolation mechanisms. However, TEE-enabled CPUs have limited computation power and memory availability, which creates unacceptable performance hurdles to run an entire model within a TEE. Linear operations (convolution, matrix multiplication, etc) are orders of magnitudes faster on a GPU compared to a TEE-enabled CPU. DarKnight offloads these compute-intensive linear operations to GPU. DarKnight's usage of TEEs is limited to protecting the privacy of data through a novel matrix masking of multiple inputs and performing non-linear operations (RelU, Maxpool).

In terms of applicability, DarKnight allows users to train using floating-point (FP) representation for model parameters, while still providing rigorous bounds on information leakage. FP models are routinely used in training due to convergence, accuracy and faster implementation considerations (Johnson, 2018; Guo et al., 2020; Imani et al., 2019). Many DNN accelerators use bfloat16 (Kalamkar et al., 2019) which is a half precision FP. This format is used in Intel Xeon Processors, Intel FP-GAs (Nervana, 2018), Google Cloud TPUs (Cloud, 2018) and Tenserflow (Google, 2018). Several prior works on protecting privacy, however, use operations on finite fields to provide formal bounds. Such an approach limits their usage to integer arithmetic on quantized models (Mohassel & Zhang, 2017; Gascón et al., 2017; So et al., 2019). In this work, we allow training to use FP values and

we bound the amount of information leakage with a rigorous mathematical proof. The information leakage is bounded by the variance of the additive noise and other parameters of the DarKnight blinding.

We implemented DarKnight using an Intel SGX-enabled CPU to perform matrix masking and non-linear DNN operations, while using an Nvidia GPU to accelerate linear operations. The blinding parameters in our experiments were chosen so as to preserve the original accuracy of training a model. Using these parameters DarKnight guarantees that no more than one bit of information is leaked from a one megapixel input image. Note that this will be an upper bound on the leaked information, assuming that the adversary has access to unlimited computation power to decode the blinded inputs. To the best of our knowledge, this is the first work that uses TEE-GPU collaboration for **training** large DNNs.

The rest of the paper is organized as follow. In Section 2, we explain the background. Section 3 describes the methodology for inference and training. In section 4 privacy theorem is provided. Experimental results are presented in section 5. In section 6, we draw the conclusion.

## 2 RELATED WORK AND BACKGROUND

### 2.1 INTEL SGX

TEEs such as ARMTrustZone (Alves, 2004), Intel SGX (Costan & Devadas, 2016), and Sanctum (Costan et al., 2016) provide an execution environment where computational integrity of user's application is guaranteed by the hardware. TEEs generally provide a limited amount of secure memory that is tamper proof even from a root user. SGX provides 128 MB as the enclave memory. An entire DNN model and data can be wrapped in an enclave for private execution but if size of the private data exceeds the 128MB TEE limit, it will pay a significant performance penalty for encryption and eviction of pages for swapping. While some types of side-channel attacks have been performed on SGX, many of these attacks are being fixed actively (Costan & Devadas, 2016; Xu et al., 2015). In this work we assume that SGX computations are invisible to the outside entities.

### 2.2 RELATED WORK

There are a variety of approaches for protecting input privacy during DNN training and inference. We categorized these approaches in Table 1. *Homomorphic encryption (HE)* techniques encrypt input data and then perform inference directly on encrypted data, albeit with significant performance penalty (and hence are rarely used in training DNNs). *Secure multi-party computing (MPC)* is another approach, where multiple non-colluding servers may use custom data exchange protocols to protect input data. However, this approach *requires multiple servers* to perform training or inference. An entirely orthogonal approach is to use *differential privacy (DiifP)*, which protects user information through probabilistic guarantees. *Additive Noise* is another approach mostly used for inference, where there is a trade-off between the privacy, computational complexity and, model accuracy. In some of the works mentioned below a combination of forenamed techniques is used. Among those approaches, (Tramer & Boneh, 2018) introduced Slalom an *inference* framework that uses TEE-GPU collaboration to protect data privacy and integrity. However, as stated in their work their quantized model was not designed for training DNNs. We elaborate on these reasons in Appendix E.

Table 1: Various prior techniques and their applicability

| | HE | MPC | TEE | DiifP | Noise |
|---|---|---|---|---|---|
| **Inference** | FHME (Gentry, 2009), MiniONN (Liu et al., 2017), CryptoNets (Gilad-Bachrach et al., 2016), Gazelle (Juvekar et al., 2018) | SGXCMP (Bahmani et al., 2017), SecureML (Mohassel & Zhang, 2017) | Mlcapsule (Hanzlik et al., 2018), ObliviousTEE (Ohrimenko et al., 2016), P-TEE (Gu et al., 2018), Slalom (Tramer & Boneh, 2018), Origami (Narra et al., 2019b) | | Arden (Wang et al., 2018), NOffload (Leroux et al., 2018), Shredder (Mireshghallah et al., 2020) |
| **Training** | | SecureML (Mohassel & Zhang, 2017), SecureNN (Wagh et al., 2019), ABY3 (Mohassel & Rindal, 2018) | MSP (Hynes et al., 2018), Chiron (Hunt et al., 2018) | DiffP (Abadi et al., 2016), Rappor (Erlingsson et al., 2014), Apple (Team, 2017) PP DNN (Shokri & Shmatikov, 2015) | |

## 3 DARKNIGHT

### 3.1 THREAT MODEL

**Adversary capabilities:** While adversaries can perform various attacks on DNN models and datasets (Riazi et al., 2019), DarKnight focuses on attacks that expose the datasets used in training or inference and attacks that modify computational results on untrusted hardware. Model privacy and side channel attacks are out of the scope of this work. Within this scope, the adversary is assumed to have the full root access to the system, which includes the GPU in our setup. The adversary cannot see any computations or data stored within the TEE. But the adversary has unrestricted access to data that leaves TEE, such as the blinded input data and can alter computational results performed

on the GPU. Since model protection is outside of the scope we assume the adversary can access the DNN model parameters.

**Information-theoretic Data Privacy:** We quantify information leakage in terms of the mutual information between original inputs and blinded inputs that are visible to the adversary. More precisely, from an information theoretical point of view, an adversary with an unlimited computation power who observes unlimited number of blinded inputs cannot gain more information about original inputs than what our upper bound on leakage provides. This upper bound itself can be controlled by the power of noise and other blinding parameters in our design. In our implementation we selected these parameters such that the overall training or inference accuracy is not reduced due to them. In section 4 and Appendix A, we provide the details of our theoretical analysis.

**Computation Integrity:** Since the adversary has access to blinded inputs, it can alter the returned values to the TEE to manipulate model training or inference. DarKnight can verify the computations performed in the unsecured GPU up to the computation precision. In the other words, DarKnight detects if the results are altered more than the computation precision by an adversary.

## 3.2 DARKNIGHT OVERVIEW

DarKnight supports both private inference and training in a single framework. Fig. 1 depicts the overall execution flow of DarKnight. A cloud server with an SGX enclave and GPU accelerator forms the computing base. DarKnight uses SGX to blind input data while enabling GPUs to perform computationally intensive linear operations on private data. The initial model (**W**) that a user wants to train is loaded into the cloud server, and is made accessible to the untrusted GPU as well. DarKnight then uses the following steps: (1) A batch of training/inference input data set is encrypted by the client using a mutually agreed keys with SGX and sent to the server. (2) SGX decrypts the images and starts the forward and backward propagation. (3) During the forward/backward pass, each layer requires some linear and nonlinear operations. Before offloading linear operations to GPU, SGX calls DarKnight's blinding mechanism to seal the data. To seal the data, DarKnight uses the notion of a *virtual batch*, where $K$ inputs are linearly combined to form $K$ coded inputs. The size of the virtual batch is limited by the size of the SGX memory that is necessary to blind $K$ images, typically 4-8 images at a time. (4) The blinded data is offloaded to GPU for linear operation. (5) GPU performs linear operations on blinded data and returns the data back to SGX labeled as step 6. (7) SGX decodes the received computational outputs using DarKnight's decoding strategy and then performs any non-linear operations within SGX. This process is repeated both for forward and backward propagation of each layer.

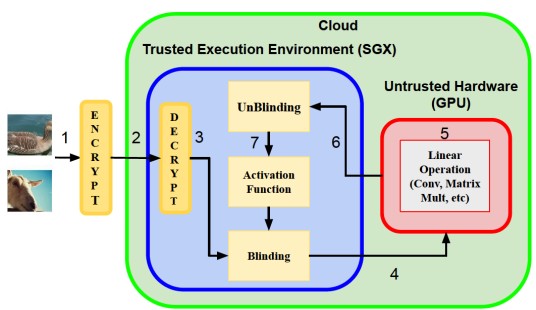

Figure 1: General steps of one forward/backward pass of DarKnight for training a DNN

## 3.3 PRIVACY IN INFERENCE

In this section, we start with DarKnight's inference strategy. We consider a trained DNN, represented by model parameters $\mathbf{W}$ with $L$ layers, which is performing inference on input $\mathbf{x}_0$, which must be protected. At a layer $l$ the inference process computes $\mathbf{y}_l = \langle \mathbf{W}_l , \mathbf{x}_l \rangle$, where $\mathbf{W}_l$ and $\mathbf{x}_l$ represent the model parameters and inputs in layer $l$, and $\langle \cdot, \cdot \rangle$ corresponds to the bilinear operation at that layer (e.g. matrix product, convolution, etc.). After the linear operation finishes, an activation function $(g(\cdot))$ creates the next layer input $\mathbf{x}_{l+1} = g(\mathbf{y}_l)$. Within this context, DarKnight first receives a set of $K$ inputs $\mathbf{x}_0^{(1)}, \ldots, \mathbf{x}_0^{(K)}$ for a batch inference from a client. Our goal is to perform linear calculations of $\mathbf{y}_0^{(1)} = \langle \mathbf{W}_0, \mathbf{x}_0^{(1)} \rangle, \ldots, \mathbf{y}_0^{(K)} = \langle \mathbf{W}_0, \mathbf{x}_0^{(K)} \rangle$ on the GPU without exposing the inputs to the GPU. Note that the subscript 0 in all these variables refers to the first layer. At this point, we drop the subscript for a more clear notation. Also, we apply $\mathbf{x}$ for the inputs that need to be protected and $\bar{\mathbf{x}}$ for the blinded inputs to visually distinguish different notations.

**Key Insight:** The main idea behind DarKnight's privacy protection scheme is the fact that the most computationally intensive operator (such as convolutions) is *bilinear*. Thus, instead of asking the GPU to calculate $\langle \mathbf{W}, \mathbf{x}^{(i)} \rangle$, which exposes the inputs, DarKnight uses matrix masking to linearly

combine the inputs and add a random noise to them. Due to the bilinear property, any linear operation on $K$ masked inputs can be recovered if there are $K$ different linear computations performed.

**Matrix Masking**: Introduced by (Cox, 1980; 1994; Kim, 1986; Spruill, 1983), matrix masking scheme can be used for variety of reasons such as noise addition, sampling and etc. The general form of $\mathbf{B}\,\mathbf{X}\,\mathbf{A} + \mathbf{C}$ is used for protecting Matrix $\mathbf{X}$. Any of these matrices can be used for masking data based on the data privacy goal. For DarKnight we use $\mathbf{A}$ and $\mathbf{C}$ as we explain the in this section.

**DarKnight Matrix Masking(Blinding)**: More specifically, DarKnight creates $K + 1$ inputs $\bar{\mathbf{x}}^{(1)}, \ldots, \bar{\mathbf{x}}^{(K)}$, as follows,

$$\bar{\mathbf{x}}^{(i)} = \alpha_{i,1}\mathbf{x}^{(1)} + \cdots + \alpha_{i,K}\mathbf{x}^{(K)} + \alpha_{i,(K+1)}\mathbf{r}, \quad i = 1, \ldots, (K+1) \tag{1}$$

The scalars $\alpha_{i,j}$, and the noise vector $\mathbf{r}$ are randomly generated; and the size of $\mathbf{r}$ matches that of $\mathbf{x}$. The scalars $\alpha_{i,j}$'s are represented by matrix $\mathbf{A}$, which are dynamically generated for each batch and securely stored inside SGX for unblinding. Hence, by revealing the values $\bar{\mathbf{x}}^{(i)}$'s to GPU, we do not expose the inputs $\mathbf{x}^{(i)}$'s. At the next step, the blinded data $\bar{\mathbf{x}}^{(i)}$'s are sent to the GPU which performs the following computations: $\bar{\mathbf{y}}^{(i)} = \langle \mathbf{W}, \bar{\mathbf{x}}^{(i)} \rangle, \quad i = 1, \ldots, (K+1)$. Please note that matrix $\mathbf{A}$ can be chosen such that its condition number close to one, so that blinding and unblinding algorithm remains numerically stable. For this purpose, orthogonal matrices serve us the best.

**DarKnight Unblinding**: The $K + 1$ outputs $\bar{\mathbf{y}}^{(i)}$ returned from the GPU must be unblinded to extract the original results $\mathbf{y}^{(i)}$. These value can be extracted as follows,

$$\bar{\mathbf{Y}} = \left\langle \mathbf{W}, [\bar{\mathbf{x}}^{(1)}, \ldots, \bar{\mathbf{x}}^{(K+1)}] \right\rangle = \underbrace{\left\langle \mathbf{W}, [\mathbf{x}^{(1)}, \ldots, \mathbf{x}^{(K)}, \mathbf{r}] \right\rangle}_{\mathbf{Y}} \cdot \mathbf{A} \Rightarrow \mathbf{Y} = \bar{\mathbf{Y}} \cdot \mathbf{A}^{-1} \tag{2}$$

**DarKnight Advantages:** (1) Unlike prior works (Tramer & Boneh, 2018) DarKnight does not need to store $\mathbf{W} \cdot \mathbf{r}$ within the SGX memory thereby significantly enhancing our ability to infer with much larger models. (2) size of the matrix $\mathbf{A}$ is proportional to the number of inputs that are blinded together (K), and is orders of magnitude smaller the model size $\mathbf{W}$. Hence, the order complexity of Blinding/Unblinding operations is much less than the linear operations ($\langle \mathbf{W}, x \rangle$) in a DNN with millions of parameters. (3) The process of unblinding $K$ inputs with one random noise requires $K + 1$ computations. During unblinding we extract $\mathbf{W} \cdot \mathbf{r}$, but that value is just dropped. Thus DarKnight trades $1/K$ additional computations in order to eliminate the need to secure very large model parameters.

### 3.4 Privacy in Training

In the training setting, for a model with $L$ layers which is being trained with a batch of $K$ inputs, the model parameters $\mathbf{W}_l$ at layer $l$ are updated using the well known SGD process as:

$$\mathbf{W}_l^{\text{new}} = \mathbf{W}_l^{\text{old}} - \eta \times \triangledown \mathbf{W}_l, \qquad \triangledown \mathbf{W}_l = \frac{1}{K} \sum_{i=1}^{K} \langle \delta_l^{(i)}, \mathbf{x}_l^{(i)} \rangle \tag{3}$$

Here $\eta$ is the learning rate, and $\delta_l^{(i)}$ is the gradient of the loss for the $i^{\text{th}}$ point in the training batch, with respect to the output of layer $l$. DarKnight must protect $\mathbf{x}_l^{(i)}$ for each layer of the DNN when the layer's linear operations are outsourced to a GPU. Recall that the decoding process for inference exploited the invariant property of model parameter for any given input such that $\left\langle \mathbf{W}, [\bar{\mathbf{x}}^{(1)}, \ldots, \bar{\mathbf{x}}^{(k+1)}] \right\rangle = \left\langle \mathbf{W}, [\mathbf{x}^{(1)}, \ldots, \mathbf{x}^{(k)}, \mathbf{r}] \right\rangle \cdot \mathbf{A}$, meaning that a single $\mathbf{W}$ was shared between all the inputs of that layers. However, during the training process, we a have different $\delta_l^{(i)}$ for each input $\mathbf{x}_l^{(i)}$. Thus, decoding the $\langle \delta_l^{(i)}, \mathbf{x}_l^{(i)} \rangle$ from obfuscated inputs $\langle \delta_l^{(i)}, \bar{\mathbf{x}}_l^{(i)} \rangle$ is a more challenging that requires training specific decoding approach.

**Key Insight:** The key insight is that while training a batch of inputs, it is not necessary to compute the $\langle \delta_l^{(i)}, \mathbf{x}_l^{(i)} \rangle$ for each input $\mathbf{x}^{(i)}$. Instead, the training process only needs to compute cumulative parameter updates for the entire batch of inputs. Hence, what is necessary to compute is the entire $\triangledown \mathbf{W}_l$ which is a summation over multiple inputs in the batch.

**DarKnight Blinding:** DarKnight exploits this insight to protect privacy without significantly increasing the blinding and unblinding complexity of the blinding process. In particular, DarKnight uses a new linear encoding scheme to combine inputs (covered by noise). As shown in equation 3,

there are $K$ inputs on which gradients are computed. Instead of calculating the $K$ products in equation 3, DarKnight calculate the following $K + 1$ equations, in the backward propagation,

$$\triangledown \mathbf{W} = \sum_{j=1}^{K+1} \gamma_j \mathrm{Eq}_j, \qquad \mathrm{Eq}_j = \left\langle \sum_{i=1}^{K} \beta_{j,i} \, \delta^{(i)} , \bar{\mathbf{x}}^{(j)} \right\rangle \tag{4}$$

In the above equations, $\bar{\mathbf{x}}^{(j)}$ is the blinded input as produced by Equation 1, while the gradients are multiplied with the $\beta_{j,i}$ in the GPU. In contrast to inference where $\mathbf{W}$'s are fixed (independent of the input), during training the parameter updates are with respect to a specific input. Hence, each $\delta_l^{(i)}$'s corresponds to different $\mathbf{x}_l^{(i)}$ during training. As such, DarKnight uses a different blinding strategy where the overall parameter updates $\triangledown \mathbf{W}$ can be decoded very efficiently. In particular, DarKnight selects $\alpha_{j,i}$'s, $\beta_{j,i}$'s and $\gamma_i$'s such that

$$\mathbf{B}^\mathsf{T} \cdot \boldsymbol{\Gamma} \cdot \mathbf{A} = \begin{bmatrix} 1 & 0 & \dots & 0 & 0 \\ 0 & 1 & 0 & \dots & 0 \\ \vdots & \ddots & \ddots & \ddots & \vdots \\ 0 & \dots & 0 & 1 & 0 \end{bmatrix}_{K \times (K+1)} \tag{5}$$

Assuming batch size is equal to $K$, the $\beta_{i,j}$ parameters used for scaling $\delta$ values is gathered in the $K + 1$ by $K$ matrix, $\mathbf{B}$. $\alpha_{i,j}$'s are gathered in the $K + 1$ by $K + 1$ matrix $\mathbf{A}$, the scalar matrix with the same size for intermediate features and $\gamma_i$'s form the diagonal of a $K + 1$ by $K + 1$ matrix $\Gamma$, that gives us the proper parameters for efficient decoding. The proof is discussed in Appendix D.

**DarKnight Unblinding:** Given the constraint imposed on $\alpha_{j,i}$'s, $\beta_{j,i}$'s and $\gamma_i$'s the decoding process is trivially simple to extract $\triangledown \mathbf{W}$. It is easy to see that if the scalars $\alpha_{i,j}$'s, $\beta_{i,j}$'s and $\gamma_i$'s satisfy the relation equation 5, we will have

$$\frac{1}{K} \sum_{j=1}^{K+1} \gamma_j \, \mathrm{Eq}_j = \frac{1}{K} \sum_{i=1}^{K} \langle \delta_l^{(i)}, \mathbf{x}_l^{(i)} \rangle = \triangledown \mathbf{W}_l \tag{6}$$

In other words, the unblinding process only involves calculating a linear combination of the values in equation 4, which are calculated in the untrusted GPU.

**DarKnight Training Complexity:** It is important to note that DarKnight's training approach for blinding and unblinding is very simple. The size of the $\alpha$, $\delta$ and $\gamma$ matrices is just proportional to the square of the batch size that is being processed at one time. Therefore, generating them for every batch has a negligible performance overhead. Even with 8-64 batch size, (commonly used in VGG training (Canziani et al., 2016; Han et al., 2015; Narra et al., 2019a) these scaling values are substantially smaller than the model parameters $\mathbf{W}$. More implementation details Appendix C.

### 3.5 EXTENDING DARKNIGHT TO VERIFY DATA INTEGRITY WITH UNTRUSTED GPU

Apart from protecting privacy, DarKnight can be extended easily to a scenario when GPU's computation cannot be trusted. In this case, the linear computations performed by the GPU must also be verified. In the interest of space, we just provide an insight into how DarKnight can perform data integrity checks for inference and we leave the details for the Appendix D. Similar extensions for training are also possible. Recall that DarKnight creates $K + 1$ blinded inputs $\bar{\mathbf{x}}^{(1)}, \dots, \bar{\mathbf{x}}^{(K+1)}$ for $K$ original inputs. To provide integrity, DarKnight creates one additional linear combination of inputs (say $\bar{\mathbf{x}}^{(K+2)}$), using the same approach as in equation 1. This additional equation allows us to verify the accuracy of each result $\mathbf{y}^{(i)}$ by computing it redundantly twice using two sets of equations. An error is detected if the difference between the two estimations is larger than our desired computation precision. In case an error is detected, TEE may perform additional corrective action, such as executing on another GPU worker or perform additional redundant computations. But these actions are outside the scope of our current work.

## 4 PRIVACY GUARANTEE

In this section, we bound the information that leaks, when using Darknight's blinding approach. In particular, we measure the amount of information the adversary can potentially gain about the raw data from the blinded data, if the adversary has access to an unlimited computation power. The amount of information leaked by $\bar{\mathbf{x}}^{(i)}$'s about $\mathbf{x}^{(j)}$ is the **mutual information (MI)** between these two variables, defined by (Cover, 1999)

$$I(\mathbf{x}^{(j)}; \bar{\mathbf{x}}^{(1)}, \dots, \bar{\mathbf{x}}^{(K+1)}) = h(\mathbf{x}^{(j)}) - h(\mathbf{x}^{(j)} | \bar{\mathbf{x}}^{(1)}, \dots, \bar{\mathbf{x}}^{(K+1)}) . \tag{7}$$

Here, $h(\cdot)$ denotes the Shannon entropy function. Note that the information that adversary can potentially learn about $\mathbf{x}^j$ by having all $\bar{\mathbf{x}}^i$'s is fundamentally bounded by $I(\mathbf{x}^{(j)}; \bar{\mathbf{x}}^{(1)}, \dots, \bar{\mathbf{x}}^{(K+1)})$. This mutual information in DarKnight can be bounded by the parameters used to blind the data.

**Theorem 1.** *Assume that $X^1, \dots, X^K$ are scalars such that $|X^i| \leq C_1$ for all $i$. Suppose $\alpha_{i,j}$'s are real non-zero scalars and $R$ denotes a Gaussian random variable with variance $\sigma^2$. Also $\bar{X}^i$ is defined as*

$$\bar{X}^i = \sum_{j=1}^{K} \alpha_{j,i} X^j + \alpha_{(K+1),i} R, \quad i = 1, \dots, K+1.\tag{8}$$

*Then the information leaked from $\bar{X}^i$'s about $X^j$ is bounded by*

$$I\left(X^j; \bar{X}^1, \dots, \bar{X}^{(K+1)}\right) \leq \frac{K^2(K+1)C_1^2 \bar{\alpha}^2}{\underline{\alpha}^2 \sigma^2}, \quad j = 1, \dots, K.\tag{9}$$

*Here $\bar{\alpha} = \max_{i,j} |\alpha_{i,j}|$ and $\underline{\alpha} = \min_{i,j} |\alpha_{i,j}|$.*

The details of our proof is provided in Appendix A. Note that there is one source of information leakage not considered in the above bound, namely the leakage of inputs from gradients with respect to weight ($\triangledown \mathbf{W}$). But as we described in Equation 4, we only provide a single model update computed across all the inputs in a batch, which is similar to the state of art secure aggregation mechanisms used to bound such leakage (Bonawitz et al., 2017; Zhu et al., 2019).

## 5 EXPERIMENTS

### 5.1 SETUP

DarKnight server consisted of an Intel(R) Coffee Lake E-2174G 3.80GHz processor and an Nvidia GeForce GTX 1080 Ti. The server has 64 GB RAM and supports Intel Soft Guard Extension (SGX). Due to enclave thread creation overheads, in our experiments a single thread was created to perform the blinding and unblinding operations within the TEE. Parts of the DarKnight inference code is based on Slalom code (Tramer & Boneh, 2018) but uses DarKnight's unique blinding an unblinding mechanisms in addition to various other enhancements, and also eliminated the need to store blinding factors within the enclave.

We used three different DNN models: VGG16 (Simonyan & Zisserman, 2014), ResNet152 (He et al., 2016) and, MobileNet (Sandler et al., 2018; Howard et al., 2017). We chose MobileNet because it is the worst-case benchmark for our model as it reduces linear operations considerably (using depth-wise separable convolution), thereby reducing the need for GPU acceleration. We used ImageNet (Russakovsky et al., 2015), CIFAR-10 and CIFAR-100 (Krizhevsky et al., 2009) as our datasets. All the parameters, models' and implementation details, and dataset descriptions are attached in the supplementary material.

### 5.2 INFERENCE RESULTS

For a fair comparison, in extracting inference timing for all of our implementations, we use the same library that Slalom used which is called Eigen library. Eigen is a high performance C++ based linear algebra library. For GPU linear operations we used Keras 2.1.5, Tenseflow 1.8.0, and Python 3.6.8.

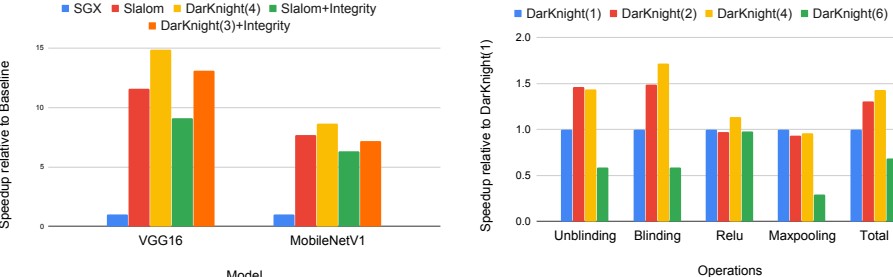

Figure 2: a)Inference speedup comparison with different implementations relative to SGX for VGG16, and MobileNetV1 on ImageNet. b) Inference speedup comparison of different operations relative to DarKnight(1) for different virtual batch-sizes for VGG16 on ImageNet.

**Inference Speedup**: Fig. 2(a) compares the speedup of the inference for VGG16 and MobileNetV1 across five different configurations. The baseline bar (SGX) performs all the calculations within SGX. The red bar uses Slalom blinding while trusting GPU that results are always correct, DarKnight(4) is our model while using a virtual batch size of 4. Slalom+Integrity bar shows the performance when Slalom's data integrity verification(Freivalds) is deployed to verify GPU computations. DarKnight(3)+Integrity uses DarKnight with virtual batch size of 3 and an additional equation to redundantly compute all the results twice for integrity verification.

For VGG16, DarKnight(4) provides $15X$ speedup, compared to the SGX only baseline, and $30\%$ improvement over Slalom. Slalom's implementation encrypts $\mathbf{W} \cdot \mathbf{r}$ and stores them outside of SGX memory. At each layer, they retrieve the necessary unblinding factors into SGX, then decrypt them before using them. When providing the additional integrity checks, DarKnight(3) provides about $13X$ speedup over baseline, and $1.45X$ speedup over Slalom. For integrity checks, we used the DarKnight(3) model in which three images are linearly combined. The reason is that when integrity checks are added to the design, we will have 5 equations and 4 unknowns. Creating an additional equation takes more SGX memory, thereby limiting the DarKnight's virtual batch size to 3, which is further quantified below. Although MobilenetV1 shows the least speedup because it reduces the number of linear operations considerably, we still have more than $8X$ speedup.

**Effect of Virtual Batch Size**: Recall that virtual batch size is the number of images that are linearly combined in equation 1. Fig. 2(b) quantifies the effect of batch size on the inference time. In the figure, DarKnight($K$) is used to denote a virtual batch size of $K$. For the same number of input data points with different batch sizes, we issue inference requests and divided the total inference time across four categories of operations: unblinding, blinding, Relu and Maxpooling operations. We used DarKnight(1) as baseline. It represents the case where a single image is combined with random Gaussian noise $r$ to create two equations using equation 1. As the virtual batch size increases the total speedup improved as long as the virtual batch size fits within SGX memory limits. As the virtual batch size exceeds 4, the execution time gets worse due to SGX memory overflow.

Table 2: Effect of different noise signals on the accuracy of DarKnight inference for VGG16, ResNet152 and MobileNetV1 on ImageNet

| Noise | VGG16 | | ResNet152 | | MobileNetV1 | | All Models |
| | Top1 Accuracy | Top5 Accuracy | Top1 Accuracy | Top5 Accuracy | Top1 Accuracy | Top5 Accuracy | **MI upper bound** |
| --- | --- | --- | --- | --- | --- | --- | --- |
| No privacy | 64.26 | 85.01 | 72.93 | 90.60 | 64.96 | 85.29 | − |
| $\mathcal{N}(4e3, 1.6e7)$ | 64.23 | 85.01 | 72.46 | 90.47 | 64.99 | 85.26 | $5 * 10^{-4}$ |
| $\mathcal{N}(1e4, 2.5e7)$ | 64.25 | 85.06 | 72.35 | 90.23 | 64.81 | 85.26 | $3.2 * 10^{-4}$ |
| $\mathcal{N}(1e4, 1e8)$ | 64.25 | 85.05 | 71.87 | 89.93 | 64.54 | 85.15 | $8 * 10^{-6}$ |
| $\mathcal{N}(0, 4e8)$ | 64.24 | 85.01 | 72.24 | 90.09 | 64.87 | 85.19 | $2 * 10^{-6}$ |
| $\mathcal{N}(0, 9e8)$ | 64.22 | 85.02 | 70.78 | 89.33 | 64.41 | 84.87 | $\mathbf{0.8 * 10^{-6}}$ |

**Mutual Information Upper Bound and Random Noise Strength**: We use a random Gaussian vector with iid entries, $\mathcal{N}(\mu, \sigma^2)$, as the noise vectors $\mathbf{r}_i$'s, where $\sigma^2$ is the order of magnitude strength over the typical model parameter values seen in a model. In Table 2, we investigated the effect of various noise strengths, on the inference accuracy. For some of the large noise strengths, a negligible accuracy loss was observed while for most cases, adding a noise signals cause no accuracy degradation. Last column represents the upper bound of mutual information. For computing that, we used the rigorous bound of Theorem 1. In this setting the value of K is set to 4, $\frac{\bar{\alpha}^2}{\alpha^2} \leq 10$ and

$C_1 \leq 1$ using $\ell_1$ normalization in prepossessing. For instance, using $\mathbf{r} = \mathcal{N}(0, 9e8)$ will bound the information leakage to $0.8 * 10^{-6}$, which is lower than one bit leakage in a Megapixel image. This selection of blinding parameters cause no accuracy loss in VGG16 and MobileNetV1, and around $2\%$ degradation in Top 1 accuracy, and $1\%$ loss in Top 5 accuracy in ResNet152.

## 5.3 TRAINING RESULTS

For SGX implementations, we used Intel Deep Neural Network Library (DNNL) for designing the DNN layers including the Convolution layer, ReLU, MaxPooling, and Eigen library for Dense layer. For linear operations on GPU we used Keras 2.1.5, Tenseflow 1.8.0, and Python 3.6.8. For evaluating training performance, two aspects are examined: accuracy impact and the execution time of training.

**Effect of Random Noise on Accuracy**: As depicted in Fig. 4, the accuracy of training for different noise strengths is measured on VGG16, ResNet152, and MobileNetV2. Fig. 4(a) shows the accuracy of training for VGG16 on CIFAR-10 dataset. The accuracy loss after epoch 50 is less than 0.001 compared to training on open data without any privacy controls. Very similar behaviour is observed across a wide range of input datasets and models. More results in Appendix D.

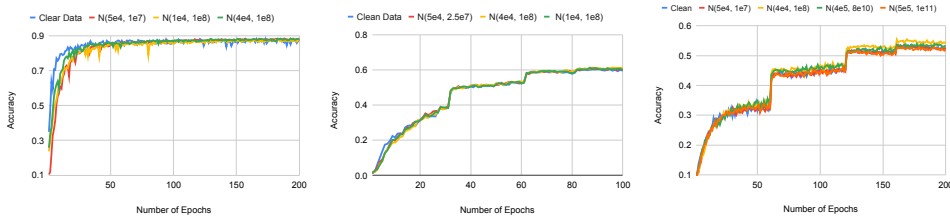

(a) CIFAR-10 on VGG16     (b) CIFAR-100 on ResNet152     (c) CIFAR-100 on MobileNetV2

Figure 3: Training accuracy of DarKnight in different DNNs and datasets for batch-size = 128

**Training Execution Time**: As illustrated in Table 3, for the baseline majority of time is spent in the linear operations ($84\%$ for VGG16). By speeding up the linear operations in DarKnight (which include convolution and matrix multiplication) balance is now tilted towards to non-linear (including ReLU, Maxpooling, blinding, unblinding, batch normalization) operations. With DarKnight the non-linear operations consume about $84\%$ of the execution time while the linear operation time is substantially reduced. Fig. 4 demonstrates the speedup of training using DarKnight relative to the baseline fully implemented on SGX. It breaks down the execution time spent into GPU operations and SGX operations. Note that SGX operations include all the non-linear operations along with the blinding and unblinding overheads. For instance for VGG16, as shown in the third sets of bar, DarKnight speeds up the total linear operation time by 52x by using the vast GPU parallelism. For SGX operations, DarKnight pays overheads for blinding/unblinding while the baseline has to deal with encryption/decryption mechanism when data does not fit the SGX memory. That is why for SGX operations we only observe 1.88 times speedup in DarKnight. Overall the execution time is improved by about $10X$ with DarKnight. As we explained MobilenetV2 reduced the amount of linear operations considerably. Even in this worst-case scenario for DarKnight, we improved the performance by 2.5 times. ResNet152 provides $4.7$ times speedup. Both ResNet and MobileNet models have batch normalization layers that are computation intensive and we cannot simply offload them to GPU. As a result their speedup is less than VGG models.

Table 3: Percentage of Execution Time Spent on Linear Operations in Training of ImageNet for VGG16, ResNet152, MobileNetV2

| Phase | VGG16 | | ResNet152 | | MobileNetV2 | |
|---|---|---|---|---|---|---|
| | DarKnight | Basline | DarKnight | Basline | DarKnight | Basline |
| Forward Pass | 0.13 | 0.90 | 0.13 | 0.62 | 0.23 | 0.50 |
| Backward propagation | 0.20 | 0.81 | 0.15 | 0.60 | 0.17 | 0.66 |
| Forward+Backward | 0.16 | 0.84 | 0.15 | 0.61 | 0.19 | 0.62 |

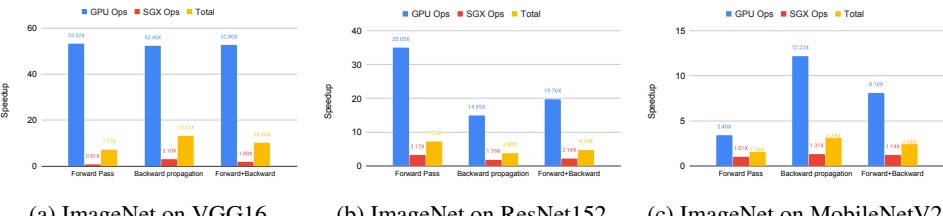

(a) ImageNet on VGG16     (b) ImageNet on ResNet152     (c) ImageNet on MobileNetV2

Figure 4: Training Execution Time Breakdown

## 6 CONCLUSION

This work proposes DarKnight a unified inference and training platform that uses TEE to perform data obfuscation and uses GPU to perform linear operations on obfuscated data. DarKnight uses a novel matrix masking to prevent data exposure. We provide a rigorous proof that bounds DarKnight's information leakage using mutual information. We achieved the privacy of 1 bit leakage on a Megapixel image while using FP operations. We evaluated three different models and datasets to demonstrate considerable speedups with provably bounded data privacy leakage and also verifying the computational integrity from GPU. For large DNNs, we observe an average of $12X$ speedup for inference and $5.8X$ speedup for training without accuracy degradation over the baseline fully implemented inside TEE.

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

## A  PRIVACY GUARANTEE

Darknight provides privacy by matrix masking. But how can one implement matrix masking, to attain desirable privacy measures?

A popular approach would be to keep all the variables in their floating point representations, while adding Gaussian noise (or uniform noise) to the vector we would like to protect. This line of work has gained attention recently, as the industry is moving towards efficient floating-point computations. Many DNN accelerators use BFloat16 (Kalamkar et al., 2019) which is a half precision floating point. This format is used in Intel Xeon Processors, Intel FPGAs (Nervana, 2018; AI, 2018), Google Cloud TPUs (Cloud, 2018; Authors, 2018) and Tenserflow (Google, 2018; Authors, 2018). In this scenario, we measure privacy through the concept of leaked information. Leaked information indicates how much information the masked vector (data after blinding) posses, about the raw data (Guo et al., 2020; Matthews et al., 2011). In the other words, it represents the amount of information the adversary can potentially gain from the raw data, *if* it has access to an unlimited computational power.

In this section, we will keep all the variables in their floating point representations. Thus, all the numbers and random variables are real valued, even when not stated explicitly. We will first explain a general matrix masking introduced by (Cox, 1980; 1994). Next, we will explain Darknight privacy, through the notation used in matrix masking. Finally, we will calculate the information leakage in our masked matrix, as a measure of privacy.

**Matrix Masking:**

Introduced by (Cox, 1980; 1994), matrix masking scheme can be used for variety of reasons such as noise addition, sampling and etc. The general form of $BXA + C$ is used for protecting Matrix X. In the above formula B, A and C are called record transformation mask, attribute transformation mask and displacing mask respectively. Any of these matrices can be used for encoding data based on the data privacy goal. For instance, (Kim, 1986) et al. first added a random noise to data and then transformed it to form a distribution with the desired expected value and variance, by carefully

tuning $A$ and $B$. (Spruill, 1983) empirically compared different masking schemes including additive and multiplicative noise. We will show that how Darknight can be a form of matrix masking, with the right choice of the matrices $A$, $B$ and $C$.

**DarKnight with Matrix Masking:**
Following our notation in equation 1, our goal is to protect the vectors $\mathbf{x}_i$, by adding a random noise to each as follows

$$\bar{\mathbf{x}}^{(i)} = \alpha_{i,1}\mathbf{x}^{(1)} + \cdots + \alpha_{i,K}\mathbf{x}^{(K)} + \alpha_{i,(K+1)}\mathbf{r} , \quad i = 1, \ldots, (K+1) , \tag{10}$$

where $\mathbf{r}$ is a random noise vector, and $\alpha_{i,j}$'s are also chosen randomly. Now first, we denote $\mathbf{X} = [\mathbf{x}^{(1)}, \ldots, \mathbf{x}^{(K)}]$ to be the matrix that we would like to protect, and $\bar{\mathbf{X}} = [\bar{\mathbf{x}}^{(1)}, \ldots, \bar{\mathbf{x}}^{(K)}]$ to be the masked matrix that we send to unsecured GPU. In this case, the equation equation 10 can be rewritten as follows.

$$\bar{\mathbf{X}} = \mathbf{X} \cdot A + \mathbf{r} \cdot \mathbf{c}^T , \tag{11}$$

where the matrix $A = [\alpha_{i,j}]_{i,j} \in \mathbb{R}^{(K+1) \times (K+1)}$ contains some values of $\alpha_{i,j}$'s, and $\mathbf{c}^T = [\alpha_{1,(K+1)}, \ldots, \alpha_{(K+1),(K+1)}]$.

We also prefer to choose a matrix A, with a condition number close to one, so that our blinding and unblinding algorithm remains numerically stable. For this purpose, orthogonal matrices serve us the best. In addition to that, the transformation of the matrix whose entities are independent and identically distributed standard normal variants is invariant under orthogonal transformations. Therefore, if an orthogonal matrix is used for blinding, the distribution of the raw data and blinded data remains the same (Kim, 1986), which is preferable in data privacy.

**Measuring Privacy Performance:**
In this section, we bound the information that leaks, when using Darknight's masking approach. The amount of information leaked by $\bar{\mathbf{x}}^{(i)}$'s about $\mathbf{x}^{(j)}$ is the **mutual information** between these two variables, defined by

$$I(\mathbf{x}^{(j)}; \bar{\mathbf{x}}^{(1)}, \ldots, \bar{\mathbf{x}}^{(K+1)}) = h(\mathbf{x}^{(j)}) - h(\mathbf{x}^{(j)}|\bar{\mathbf{x}}^{(1)}, \ldots, \bar{\mathbf{x}}^{(K+1)}) . \tag{12}$$

Here, $h(\cdot)$ denotes the Shannon entropy function. Note that the information that adversary can potentially learn about $\mathbf{x}^j$ by having all $\bar{\mathbf{x}}^i$'s is fundamentally bounded by $I(\mathbf{x}^{(j)}; \bar{\mathbf{x}}^{(1)}, \ldots, \bar{\mathbf{x}}^{(K+1)})$. Next, we will rigorously bound this information leakage and show how it can be bounded by properties of the noise.

**Theorem 2.** *Assume that $X^1, \ldots, X^K$ are scalars such that $|X^i| \leq C_1$ for all $i$. Suppose $\alpha_{i,j}$'s are real non-zero scalars and $R$ denotes a Gaussian random variable with variance $\sigma^2$. Also $\bar{X}^i$ is defined as*

$$\bar{X}^i = \sum_{j=1}^{K} \alpha_{j,i}X^j + \alpha_{(K+1),i}R , \quad i = 1, \ldots, K+1 . \tag{13}$$

*Then the information leaked from $\bar{X}^i$'s about $X^j$ is bounded by*

$$I\left(X^j; \bar{X}^1, \ldots, \bar{X}^{(K+1)}\right) \leq \frac{K^2(K+1)C_1^2\bar{\alpha}^2}{\underline{\alpha}^2\sigma^2} , \quad j = 1, \ldots, K . \tag{14}$$

*Here $\bar{\alpha} = \max_{i,j}|\alpha_{i,j}|$ and $\underline{\alpha} = \min_{i,j}|\alpha_{i,j}|$.*

*Proof.* To prove equation 14, first note that

$$I\left(X^j; \bar{X}^1, \ldots, \bar{X}^{(K+1)}\right) \leq \sum_{i=1}^{K+1} I\left(X^j; \bar{X}^i\right) \leq (K+1) \cdot \max_i I\left(X^j; \bar{X}^i\right) . \tag{15}$$

Now if we conclude the proof by just showing that $I(X^j; \bar{X}^i) \leq \frac{K^2 \, C_1^2 \bar{\alpha}^2}{\alpha^2 \sigma^2}$ .Since $\alpha_{i,j}$'s are non-zero, we have

$$
\begin{aligned}
I\left(X^j; \bar{X}^i\right) &= I\left(\alpha_{j,i} X^j; \bar{X}^i\right) \\
&\overset{(1)}{=} I\left(\alpha_{j,i} X^j; \sum_{l=1}^{K} \alpha_{l,i} X^l + \alpha_{(K+1),i} R\right) \\
&\overset{(2)}{=} H\left(\sum_{l=1}^{K} \alpha_{l,i} X^l + \alpha_{(K+1),i} R\right) - H\left(\sum_{\substack{l=1 \\ l \neq j}}^{K} \alpha_{l,i} X^l + \alpha_{(K+1),i} R\right) \\
&\overset{(3)}{\leq} H\left(\sum_{l=1}^{K} \alpha_{l,i} X^l + \alpha_{(K+1),i} R\right) - H\left(\alpha_{(K+1),i} R\right) \\
&= I\left(\sum_{l=1}^{K} \alpha_{l,i} X^l; \sum_{l=1}^{K} \alpha_{l,i} X^l + \alpha_{(K+1),i} R\right) .
\end{aligned}
\tag{16}
$$

Here, for equality (1), we simply replaces $\bar{X}^i$ with its definition. (2) is due to the definition of the mutual information ( $I(X; X+Y) = H(X+Y) - H(Y)$). Finally, inequality (3) holds due to Lemma 1.

Now, note that since $|X^l| \leq C_1$, we have

$$
\left| \sum_{l=1}^{K} \alpha_{l,i} X^l \right| \leq C_1 \sum_{l=1}^{K} |\alpha_{l,i}| \leq K \, C_1 \, \max_{l,i} |\alpha_{l,i}|
\tag{17}
$$

Thus, $\sum_{l=1}^{K} \alpha_{l,i} X^l$ is bounded by $K C_1 \bar{\alpha}$ and also $\alpha_{(K+1),i} R$ is a zero-mean Gaussian random variable with variance $\alpha_{(K+1),i}^2 \sigma^2$. Therefore, using Lemma 2, we have

$$
\begin{aligned}
I\left(\sum_{l=1}^{K} \alpha_{l,i} X^l; \sum_{l=1}^{K} \alpha_{l,i} X^l + \alpha_{(K+1),i} R\right) &\leq \frac{\mathrm{Var}\left(\sum_{l=1}^{K} \alpha_{l,i} X^l\right)}{\alpha_{(K+1),i}^2 \sigma^2} \\
&\leq \frac{K^2 C_1^2 \bar{\alpha}^2}{\alpha^2 \sigma^2}
\end{aligned}
\tag{18}
$$

Finally, using equation 16, equation 18, we conclude that

$$
I\left(X^j; \bar{X}^i\right) \leq \frac{K^2 C_1^2 \bar{\alpha}^2}{\alpha^2 \sigma^2}
\tag{19}
$$

Combining equation 14 and equation 19, yields our result,

$$
I\left(X^j; \bar{X}^1, \ldots, \bar{X}^{(K+1)}\right) \leq (K+1) \cdot \max_i I\left(X^j; \bar{X}^i\right) \leq \frac{K^2 (K+1) C_1^2 \bar{\alpha}^2}{\alpha^2 \sigma^2}
\tag{20}
$$

$\square$

**Lemma 1.** *Suppose that $X$ and $Y$ are two independent random variables. Then we have,*

$$
\max\{H(X), H(Y)\} \leq H(X+Y) .
\tag{21}
$$

*Proof.* Since $X$ and $Y$ are independent, we have $H(X+Y|X) = H(Y|X)$ and $H(Y|X) = H(Y)$. Therefore,

$$
H(X+Y) \geq H(X+Y|X) = H(Y|X) = H(Y) .
\tag{22}
$$

The same argument shows that $H(X+Y) \geq H(X)$, which concludes the proof. $\square$

**Lemma 2.** *Assume that $X \sim P_X$ is a random variable, and $R \sim \mathcal{N}(\mu, \sigma^2)$ is a Gaussian random variable with variance $\sigma^2$ and mean $\mu$. Then we have,*

$$I(X; X + R) \leq \frac{Var(X)}{\sigma^2} , \tag{23}$$

*where $Var(X)$ is variance of the random variable $X$.*

*Proof.* Because $\mu$ is fixed and so $I(X; X + R) = I(X; X + R - \mu)$, without loss of generality, we assume that $\mu = 0$. Define the random variable $Z$ to be $Z = X + R$, and let $f_X(\cdot)$ and $f_Z(\cdot)$ to be the probability density function of the random variables $X$ and $Z$, respectively. Also let $\phi(x) = \frac{1}{\sqrt{2\pi}} e^{-x^2/2}$ to be the probability density function of a standard Gaussian random variable. We are interested in

$$I(X; X + R) = I(X; Z) = H(Z) - H(R) . \tag{24}$$

Since $R$ is a zero-mean Gaussian random variables with variance $\sigma^2$, we have

$$H(R) = \frac{1}{2} \log(2\pi e \sigma^2) . \tag{25}$$

It remains to bound $H(z)$ in equation 24. Note that the probability distribution function of $Z$, $f_Z(\cdot)$, will be

$$f_Z(t) = \int_{-\infty}^{\infty} f_X(x) \frac{1}{\sigma} \phi\left(\frac{x-t}{\sigma}\right) dx = \mathbb{E}_{x \sim P_X}\left[\frac{1}{\sigma} \phi\left(\frac{x-t}{\sigma}\right)\right] , \tag{26}$$

where the last expected value is over the distribution of $X$. Now that we have pdf of the random variable $Z$, we can calculate and bound its entropy as follows,

$$
\begin{aligned}
H(Z) &= -\int f_Z(z) \log\left(f_Z(z)\right) \, dz \\
&= -\int \mathbb{E}_{x_1 \sim P_X}\left[\frac{1}{\sigma}\phi\left(\frac{x_1-z}{\sigma}\right)\right] \log\left(\mathbb{E}_{x_2 \sim P_X}\left[\frac{1}{\sigma}\phi\left(\frac{x_2-z}{\sigma}\right)\right]\right) \, dz \\
&\leq -\int \mathbb{E}_{x_1 \sim P_X}\left[\frac{1}{\sigma}\phi\left(\frac{x_1-z}{\sigma}\right)\right] \mathbb{E}_{x_2 \sim P_X}\left[\log\left(\frac{1}{\sigma}\phi\left(\frac{x_2-z}{\sigma}\right)\right)\right] \, dz \\
&= \mathbb{E}_{x_1, x_2 \sim P_X}\left[-\int \frac{1}{\sigma}\phi\left(\frac{x_1-z}{\sigma}\right) \log\left(\frac{1}{\sigma}\phi\left(\frac{x_2-z}{\sigma}\right)\right) \, dz\right] .
\end{aligned} \tag{27}
$$

The only inequality in the equations above is due to Jensen's inequality (since $-\log(x)$ is a convex function). Now we can explicitly calculate the integral in the last line of equation 27 (as they are all Gaussian integrals).

$$-\int \frac{1}{\sigma}\phi\left(\frac{x_1-z}{\sigma}\right) \log\left(\frac{1}{\sigma}\phi\left(\frac{x_2-z}{\sigma}\right)\right) \, dz = \frac{1}{2}\log(2\pi e\sigma^2) + \frac{1}{2\sigma^2}(x_1 - x_2)^2 . \tag{28}$$

Combining equation 28 and equation 27 bounds $H(Z)$ as desired,

$$H(Z) \leq \mathbb{E}_{x_1, x_2 \sim P_X}\left[\frac{1}{2}\log(2\pi e\sigma^2) + \frac{1}{2\sigma^2}(x_1 - x_2)^2\right] = \frac{1}{2}\log(2\pi e\sigma^2) + \frac{Var(X)}{\sigma^2} . \tag{29}$$

Finally, combining equation 24, equation 25 and equation 29 yields

$$I(X; X + R) \leq \frac{1}{2}\log(2\pi e\sigma^2) + \frac{Var(X)}{\sigma^2} - \frac{1}{2}\log(2\pi e\sigma^2) = \frac{Var(X)}{\sigma^2} , \tag{30}$$

which concludes the proof. $\qquad\square$

Theorem 2 shows that by increasing power of the noise, one can arbitrarily reduce the leaked information. Please note that for deep learning applications normalization is common in the prepossessing phase. Furthermore, many of the networks such as MobileNet and ResNet variants take advantage of the batch normalization layers. Hence, the value of $C_1$ in the above theorem is bound by $N^{(\frac{-1}{2})}$ in case $\ell_2$ normalization is used (which obviously implies $C_1 \leq 1$). With a batch size of K = 4 in the our scheme, setting variance of the noise, **r**, to be $\sigma^2 = 8e^8$, and limiting $\frac{\bar{\alpha}^2}{\alpha^2} < 10$, we have the upper bound of $1e-6$ on the leaked information, through Theorem 2. This means that in a Megapixel image, the maximum information leakage is bound by 1 pixel.

# B    Training Decoding

$$
\mathbf{A} = \begin{bmatrix} \alpha_{1,1} & \cdots & \alpha_{1,K+1} \\ \vdots & \ddots & \vdots \\ \alpha_{K+1,1} & \cdots & \alpha_{K+1,K+1} \end{bmatrix}_{(K+1)\times(K+1)} , \mathbf{X} = \begin{bmatrix} x_1^{(1)} & \cdots & x_1^{(K)} & \mathbf{r}_1 \\ x_1^{(1)} & \cdots & x_1^{(K)} & \mathbf{r}_2 \\ \vdots & \vdots & \vdots & \vdots \\ x_N^{(K)} & \cdots & x_N^{(K)} & \mathbf{r}_N \end{bmatrix}_{(N)\times(K+1)} ,
$$

$$
\mathbf{B} = \begin{bmatrix} \beta_{1,1} & \cdots & \beta_{1,K} \\ \vdots & \ddots & \vdots \\ \beta_{2,1} & \cdots & \beta_{K,K+1} \end{bmatrix}_{(K+1)\times(K)} , \delta = \begin{bmatrix} \delta_1^{(1)} & \cdots & \delta_1^{(K)} \\ \delta_2^{(1)} & \cdots & \delta_2^{(K)} \\ \vdots & \vdots & \vdots \\ \delta_N^{(K)} & \cdots & \delta_N^{(K)} \end{bmatrix}_{(N)\times(K)} ,
$$

$$
\bar{\mathbf{X}} = \begin{bmatrix} \bar{x}_1^{(1)} & \cdots & \bar{x}_1^{(K+1)} \\ \bar{x}_2^{(1)} & \cdots & \bar{x}_2^{(K+1)} \\ \vdots & \vdots & \vdots \\ \bar{x}_N^{(1)} & \bar{x}_N^{(2)} & \bar{x}_N^{(K+1)} \end{bmatrix}_{(N)\times(K+1)} , \quad \bar{\delta} = \begin{bmatrix} \bar{\delta}_1^{(1)} & \cdots & \bar{\delta}_1^{(K)} \\ \bar{\delta}_2^{(1)} & \cdots & \bar{\delta}_2^{(K)} \\ \vdots & \vdots & \vdots \\ \bar{\delta}_N^{(1)} & \cdots & \bar{\delta}_N^{(K)} \end{bmatrix}_{(N)\times(K)}
$$

We formed $\bar{\mathbf{x}} = \mathbf{x}\mathbf{A}^T$ in TEE and send it to GPU, at the same time GPU generates $\bar{\delta} = \delta\mathbf{B}^T$ and then computes $\bar{\mathbf{y}} = \bar{\delta}^T\bar{\mathbf{x}}$.

$$
Tr[\mathbf{X}] = \sum_{i=1}^{K} \mathbf{X_{ii}}
$$
$$
Tr[\mathbf{XY}] = Tr[\mathbf{YX}]
$$
$$
(\mathbf{AB})^T = \mathbf{B}^T\mathbf{A}^T
$$
$$
(\mathbf{A}^T)^T = \mathbf{A}
$$

The goal is to compute $\sum_{i=1}^{K} \delta^{(i)}\mathbf{x}^{(i)}$ by simply multiplying

$$
\mathrm{Tr}(\mathbf{\Gamma}\bar{\delta}^T\bar{\mathbf{X}}) = \mathrm{Tr}(\mathbf{\Gamma}\mathbf{B}\delta^T\mathbf{x}\mathbf{A}^T) = \mathrm{Tr}(\mathbf{A}^T\mathbf{\Gamma}\mathbf{B}\delta^T\mathbf{x}) = \mathrm{Tr}((\mathbf{B}^T\mathbf{\Gamma}\mathbf{A})^T.(\delta^T\mathbf{x}))
$$

Hence if the below equation holds, we get the desired combination in the decoding:

$$
\mathbf{B}^{\mathsf{T}} \cdot \mathbf{\Gamma} \cdot \mathbf{A} = \begin{bmatrix} 1 & 0 & \cdots & 0 & 0 \\ 0 & 1 & 0 & \cdots & 0 \\ \vdots & \ddots & \ddots & \ddots & \vdots \\ 0 & \cdots & 0 & 1 & 0 \end{bmatrix}_{K\times(K+1)}
$$

# C    Training Algorithm

In this section we provide implementation modifications for training algorithm.

**Forward Pass**: In order to use intermediate feature maps for backward propagation, we need to store them during forward pass. To achieve this goal without compromising privacy, the encoded inputs($\bar{\mathbf{x}}^{(i)}$) are stored in GPU memory during forward pass.

**Backward Propagation**: In the backward propagation there are two sets of linear operations that we are willing to offload to GPU. First is computing gradients with respect to input ($\delta$) and second is the gradient with respect to weight ($\nabla\mathbf{W}$). For computing ($\delta$), wights are already stored in the GPU, derivative of activation function is computed inside SGX and offloaded to GPU, $\delta$ of the next layer is also computed in the GPU and passed to this layer. on the other hand, for computing ($\nabla\mathbf{W}$) the blinded inputs are used. We also do not expose each individuals weight update as we explain in the weight update section.

**Matrix B**: Please note that in equation 4, the encoding of $\delta$ does not have a privacy reasons. We use this type of encoding for enabling the simple secure aggregation mechanism. In other words, linearly combining $\delta^{(i)}$s can be operated in the GPU and we do not need to protect matrix **B**.

**Matrix A and $\Gamma$**: We need to mention In equation 5, the only exposed matrix is matrix **B**. Matrix **A** and $\Gamma$ are kept inside SGX and we use them for blinding/unblinding.

**Weight Updates and DarKnight Secure Aggregation**: We use a customized version of secure aggregation for weigh updates (Bonawitz et al., 2017; Zhu et al., 2019). As explained in section 3.4, weights are stored outside the enclave and hence, we need to send $\triangledown\mathbf{W}$ to the untrusted hardware to update the weights. To protect the input data, we need to have in mind $\triangledown\mathbf{W}$ may leak some information about the intermediate features which leads to input leakage. For preventing this data leakage, one solution is increasing the batch size and, hence the number of $\triangledown\mathbf{W}^i$'s that are merged increases. Nonetheless, it is infeasible to store $\triangledown\mathbf{W}^i$ for all the layers inside SGX at the same time because of the memory limitation. To resolve this, we introduce the term *Virtual Batch*, which is essentially the largest number of images that we can be processed at the same time and fits inside SGX. After each virtual batch computation, we have to evict the pages storing $\triangledown\mathbf{W}$ to the untrusted hardware considering required precautions to store encrypted data. After computing a certain number of $\triangledown\mathbf{W}^i$'s, we use a sharding mechanism and partition these $\triangledown\mathbf{W}^i$'s, read the corresponding partitions inside the SGX, process them one by one and send the updates to the weights on the fly.

The algorithm backward propagation for one epoch is shown below:

---
**Algorithm 1** Backward Propagation Algorithm for one epoch

---
1:  **procedure** BACKWARD($\mathbf{W},\mathcal{L}$)                    ▷ gets current weights and loss function as inputs
2:      $V \leftarrow VirtualBatch.size()$
3:      $N \leftarrow \frac{K}{V}$
4:      **for** $i = 1, 2, \ldots, N$ **do**                                      ▷ for each virtual batch
5:          Pages = []
6:          **for** $l = 1, 2, \ldots, L$ **do**                                  ▷ for each layer
7:              Compute $\triangledown\mathbf{W}_l^v$                ▷ weight updates for layer l and virtual batch v
8:              Compute $\delta_{l-1}^v$                        ▷ delta of the previous layer for virtual batch v
9:          Encrypt($Page_v$)                                        ▷ Encrypt page v for a virtual batch v
10:         Evict($Page_v$)                                              ▷ Evict page for virtual batch v
11:         Pages.append($\&Page_v$)                                    ▷ Keeps the pointer to $Page_v$)
12:         $\triangledown\mathbf{W}_l \leftarrow$ Weight_Update(Pages)
13:         $\mathbf{W}_l^{\text{new}} = \mathbf{W}_l^{\text{new}} - \eta \times \triangledown\mathbf{W}_l$
14:         **return** $w$                                    ▷ the new weights for the next epoch

---

In line 3, the actual batch size is divided to the virtual batch size to get the number of virtual batches we have to process. Line 4 repeats the For loop for each virtual batch size. Line 6 to 8 shows how $\triangledown\mathbf{W}$ is computed for each layer of DNN Line 9 Encrypt page $v$ that is containing all the weight updates of the network for that virtual batch. Line 10 calls evict function for page v that is containing $\triangledown\mathbf{W}^v$. As a result, this page is moved to GPU memory with all the precaution. In line 11 that a reference to that specific page will be saved for our further computations. Line 12 calls $Weight\_Update$ function. This function has a reference to all the pages containing part of the $\triangledown\mathbf{W}$ and construct the whole $\triangledown\mathbf{W}$. Finally in Line 13 the $\triangledown\mathbf{W}$ is sent to GPU for the weight updates.

**Training Accuracy**: More results on the effect of noise power on training accuracy is provided in Fig. 5.

## D   INTEGRITY

As explained in section 3.5, we provide integrity check by adding a redundant equation to each virtual batch. This leads to having $K + 2$ linear equations for recovering $K + 1$ unknowns. The extra equation can help us verify the solution of the first $K + 1$ equation for the $K + 1$ unknowns, in the last equation. If the solution is not consistent with the last equations, this means one of the GPU cores may not function properly or their data is modified by an attacker. So any single faulty

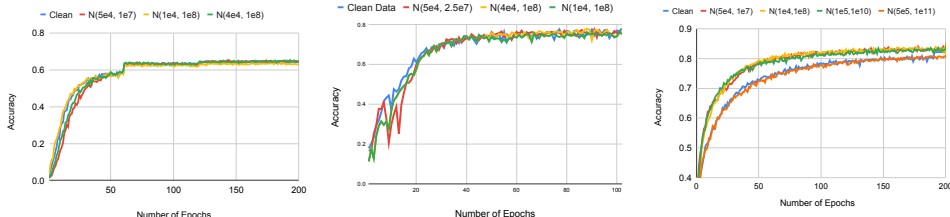

| (a) CIFAR-100 on VGG16 | (b) CIFAR-10 on ResNet152 | (c) CIFAR-10 on MobileNetV2 |

Figure 5: Training accuracy of DarKnight in different DNNs and datasets for batch-size = 128

computation or multiple non-colluding faulty computations can be detected. The forenamed system of linear equations is described below.

$$\nabla \mathbf{W}_l = \sum_{j=1}^{K+1} \gamma_j \mathbf{Eq}_j, \qquad \mathbf{Eq}_j = \left\langle \sum_{i=1}^{K} \alpha_{j,i} \, \delta_l^{(i)} \, , \, \sum_{i=1}^{K} (\beta_{j,i} \, \mathbf{x}_l^{(i)}) + \beta_{j,(K+1)} \, \mathbf{r} \right\rangle \qquad j = 1, \ldots K+2$$

(31)

In order to avoid floating point arithmetic round off errors affecting our judgment on error detection, we define a threshold on the amount of data mismatch our system can tolerate. This threshold should be larger than the system precision because we do not want computation round off to be flagged as error. In a scenario where an adversary adds a perturbation to modify the result, If the amount of this perturbation is less than the system precision, it doesn't affect the convergence and hence we can discard it.

## E  SLALOM

Among those approaches, Tramer & Boneh (2018) introduced Slalom an inference framework that uses TEEs to protect data privacy and integrity. Slalom uses the Intel SGX enclave to blind input data $\mathbf{x}$ from a client with an additive stream cipher noise $\mathbf{r}$. The blinded data $(\mathbf{x} + \mathbf{r})$ is then sent to an untrusted GPU where linear operations are performed. The computed data $\mathbf{W} \cdot (\mathbf{x} + \mathbf{r})$ is then returned to the enclave which can decode the correct computational output $\mathbf{W} \cdot \mathbf{x}$ by subtracting the precomputed $\mathbf{W} \cdot \mathbf{r}$. Here $\mathbf{W}$ is the model parameter matrix. Securely storing multiple instances of $\mathbf{r}$'s and their corresponding $\mathbf{W} \cdot \mathbf{r}$'s within the enclave memory, occupies a substantial amount of memory for large DNNs. On the other hand, storing an encrypted version of these values outside the enclave memory, leads to significant encryption and decryption costs, as these values are needed after each linear operation. In addition to that, Slalom cannot be used for training, since it precomputes $\mathbf{W} \cdot \mathbf{r}$. Precomputing the blinding factors is not feasible during training since the model parameters $\mathbf{W}$ are updated after processing every batch. Computing $\mathbf{W} \cdot \mathbf{r}$ inside the SGX after every batch also defeats the purpose of offloading the linear computations to GPU. Moreover, Slalom works on quantized models which needs a meticulous quantization algorithm not to affect the accuracy of training. Hence, the idea cannot be used in training as it is. Our idea addressed all the issues that Slalom paper introduced as its challenges for training.

## F  GPU COMPARISON AND SCALABILITY

Table 4 shows the performance of a baseline that uses GPU to perform the entire training. Clearly users of this baseline cannot protect their data. However, using a GPU for training large DNNs can give us 120X speedup relative to baseline fully implemented on SGX. DarKnight bridges this gap by more than 10X while protecting data privacy. Hence, there is no free lunch to obtain privacy, but our goal is to provide the best performance when privacy is not optional, as is the case in many operational settings such as medical imaging.

As we showed in our earlier analysis DarKnight tilts the computational balance from linear operations to non-linear operations running on SGX. Hence, the overall execution time is determined by SGX performance. But this tradeoff is not a fundamental bottleneck to scalability. With the shift towards server disaggregation Lim et al. (2009); Guleria et al. (2019) in the cloud it is possible to separate GPU pool from CPU pool and increase the number of concurrent SGX enclaves that

Table 4: Speedup in GPU relative to SGX in VGG16 Training on ImageNet. The baseline is implemented fully on Intel SGX

| Operations | Linear Ops | Maxpool Time | Relu Time | Total |
|---|---|---|---|---|
| Forward Pass | 126.85 | 11.86 | 119.60 | 119.03 |
| Backward Propagation | 149.13 | 5.47 | 6.59 | 124.56 |

can concurrently feed a GPU. Furthermore, DarKnight can be seamlessly adapted to a distributed training platform that uses data parallelism across GPUs, since inputs sent to different GPUs can be concurrently blinded using a dedicated set of SGX enclaves that feed these GPUs. Hence, we believe DarKnight has cost-effective scalability characteristics when privacy is an important consideration in training.

