# OpenReview forum: "DarKnight: A Data Privacy Scheme for Training and Inference of Deep Neural Networks"
_ICLR.cc/2021/Conference — Reject_

### Official Review · AnonReviewer3 · 2020-10-27

**Rating:** 5
**Confidence:** 4

**Review:**

The paper uses a combination of a Trusted Execution Environment and masking techniques to perform secure inference and training with the help of GPUs.

Overall the paper is well written but could be improved in the presentation of its security guarantees and how it presents related work.

Table 1 is confusing; please consider removing it or clarifying that security/privacy guarantees in each approach are completely different and solve orthogonal problems. Additionally please consider referencing the work more precisely on what training and inference algorithms you refer to.
For example:
- how do DP mechanisms such as Rappor perform training
- which inference algorithms do you refer to in MlCapsule, which training algorithms in ObliviousTEE.
- is Chiron used only for training.

Privacy/integrity guarantees obtained should be clearly stated; how they are different from those achieved by SGX, FHE, MPC, DiffPrivacy. Please see table 1 in Slalom (ICLR 2019) for an example.
It also seems that weights are revealed after every batch update. The paper says that it is fine referring to the work by Bonawitz et al.; but they do so in federated learning setting which is different from the one considered here.

Clarity:
The paper is presented in an accessible manner.
Some details however need to be explained further regarding generation of the masking variables for inference and training. For example, how are \alpha, \beta and \gamma generated. It is also important to include the time it takes to compute them in the evaluation. It seems they have to be re-generated for every batch.
In the experimental section please clarify if blinded batches can be generated in a streaming manner.


Originality:
The paper expands the ideas from Slalom (ICLR 2019) of outsourcing some computations from SGX to GPU to speed them up. While Slalom considered only inference, this work considers training as well. The masking technique is different from Slalom and gives a performance overhead in the number of input/batch parameters as opposed to the size of the network. But it seems, that integrity and privacy that the authors obtain in return is different.

Significance:
The paper tries to overcome some of the issues of using TEEs for ML — CPU and small memory sizes — by offloading linear computations to GPUs. The evaluation shows that it does obtain speedups compared to pure SGX baseline, that is, blinding/unblinding is still faster than linear operations. Hence, the work is important. However, privacy and integrity guarantees are weaker than what one will obtain with Slalom (for inference) and SGX.


pros:
- an interesting idea to mask only the inputs and not the weights during inference and training
- speedup from outsourcing linear operations to GPUs from SGX seems worth paying the cost of masking

cons:
- overall security guarantees are not formally/informally concisely stated
- integrity and privacy guarantees are weaker than related work
- it is not clear these guarantees will be desirable in practice

Spelling:
- Basline
- DiiP
- page 4 colours seem to change from blue to red when referring to the same variables.
- Intel Soft Guard Extension
- we simply replaces

---

> ### Author Response · Authors · 2020-11-14
> **Suggestions will be addressed**
>
> Thank you very much for your detailed review.
>
> Prior works: As you mentioned the privacy guarantee of the methods in Table 1 is different and we can address that in the final version of the paper. The table was an overview of the SoA methods of different mechanisms to provide a quick visual due to space limitations.
>
> Weight updates: While we agree that the federated ML approach is different than DarKnight’s focus, the similarity lies in the fact both approaches do not want individual weight updates to compromise input privacy. In this regard, they share a common goal. DarKnight’s approach to protecting information leakage is to prevent individual weight updates from each image being visible to the GPU (or other adversaries).  As shown in Eq.6 in the paper DarKnight only discloses a collective weight update across multiple inputs. As explained in Appendix C, DarKnight can merge even larger groups of inputs into a single weight update before sending it outside of SGX.
>
> Clarity: Please note that random scalars are generated dynamically for each equation generated inside SGX. The random number generation time is negligible and it can be overlapped with GPU computation time: when GPUs do the linear computations, SGX can generate those scalars. We can add this result to our final version of the work. Also, see above our general remarks regarding privacy guarantees and the necessity of floating-point values in training.

---

### Official Review · AnonReviewer2 · 2020-10-28
**Secure training and inference of DNNs, guarantees are a bit shaky**

**Rating:** 5
**Confidence:** 3

**Review:**

Summary:
This paper aims at addressing both  secure training and inference of DNNs. The proposed method relies on securely off-loading the compute-intensive part of the operations from a trusted CPU environment which has low-performance, to an untrusted high-performance gpu. Their suggested method builds on SOTA, Slalom, but is shown to have less memory over-head. Also, they point-out that their proposed scheme supports training phase as well, as opposed to Slalom which targets inference. For off-loading the computation to the untrusted GPU they propose some matrix blinding techniques.


pros:

+ The problem is interesting, and moving towards a solution for it is really helpful, since right now trusted environments only exist for CPUs in practice, which hinders fast execution of matmuls, as done in GPUs.

+ The approach seems to offer some speedup over the SOTA, Slalom. However the provided guarantees seem questionable.


cons:

- About the training phase, I think the nuances of security and privacy are a bit tangled here. Private training refers to protecting the training data, so that sensitive information within the data is not encoded/embedded in the model once the training is finished. This would be similar to what differential privacy and differentially private SGD provide. However, the suggested method in the paper seems to aim at providing secure training, in which data is not exposed to outsiders during training, however private information might still leak to the model, based on my understanding. I would like to know if that is the case.

- I don't quite understand that during training, what is sent to the GPU and what is calculated locally in the trusted environment? Since the GPU needs the weights in the clear for the forward pass, and calculates the gradients for backprop, it seems to have access to the gradients as well, which can be subject to attacks, shown in [1].

- The privacy guarantees are not completely clear to me.  It seems that the work is built upon the notion of Mutual Information, a heuristic approach. However, since this notion is average case, what would that mean for the guarantees? How does it effect each data point? It is stated that " Using these parameters DarKnight guarantees that no more than one bit of information is
leaked from a one megapixel input image. ", is this on average? or for each image, the guarantee is that no more than one bit leaks?


References

[1] Zhu L, Liu Z, Han S. Deep leakage from gradients. InAdvances in Neural Information Processing Systems 2019 (pp. 14774-14784).

---

> ### Author Response · Authors · 2020-11-14
> **Clarification on Training**
>
> Thank you for the review and we are glad that you find the work interesting.
>
> Privacy and gradient leakage: As we mentioned in the body of the paper, we provide data privacy not model privacy; Model privacy is an interesting topic for our future works. The Deep leakage paper you suggested is already cited in our submission and we in fact considered how gradients may leak inputs. As stated in the Deep leakage paper “ increasing the batch size makes the leakage more difficult because there are more variables to solve during optimization”.  Hence, the Deep leakage paper explicitly states that information leakage becomes very hard to expose when using larger batch sizes. That is precisely why DarKnight does NOT disclose any individual image gradient. Instead, it only exposes the cumulative gradient computed over a batch of inputs as has been shown in Equation 6. In addition to that, as explained in Appendix C, DarKnight can merge the weight updates for multiple batches further eliminating the information leakage before sending it to GPU.
>
> MI: Mutual information is an information-theoretic metric that has a sound basis for privacy quantification. The formula measures the mutual information between one pixel in the original image and corresponding pixels of encoded images. Hence, it is a rigorous upper bound and it is not on average over several pixels, since it is measured for each pixel in our paper.
>
> I hope this addresses your concern about the privacy of training.

---

### Official Review · AnonReviewer4 · 2020-10-29
**Some concerns about the correctness of the claims**

**Rating:** 3
**Confidence:** 3

**Review:**

Paper Summary:

The paper builds on previous work like Slalom to propose a new secure training and inference protocol in the TEE+GPU paradigm. The main technical contribution of this work is a new blinding algorithm that dramatically reduces the memory required to store the blinding parameters (decoupling it from the input/model size). The authors then build on this to extend their blinding scheme to the training use-case.

Score Rationale:

- The proposed blinding scheme does indeed provide a ~1.5x performance improvement over the Slalom baseline
- There two concerns about the correctness of the security argument provided by the authors
  - The security proof as argued in the paper does not extend from the single pixel to multi-pixel case and as such does not apply to real-world images
  - The reviewer believes that this not a mere gap in the security proof. Infact for certain allowed settings of parameters there are practical attacks.

Detailed Comments:

- The core security argument is rooted in Theorem 1. This theorem in prose rightly claims that the blinded pixels at any specific index, leak almost no information about the pixels at that specifc index in the source images.
- This statement critically makes no claims whether all the pixels in all the blinded images in their totality will leak information about a given pixel index in the source images.
- The authors then argue that the total information leaked is bounded by the sum of the information leaked at each index (based on the blinded pixels at that particular index).
- The statement is not true. The total information leaked is bounded by the sum of the information leaked at each index (based on the blinded pixels across all indices)
- The next concern to evaluate is whether this is just a technical/theoretical gap in the proof or whether it leads to a practical attack.
- Consider the case where the parameters are set as in section 5.2 with K=1. Note that reducing K improves the leakage bound in principle.
- In this particular case we can rewrite the blinding equations s.t. $x^{(1)} = \frac{\overline{\mathbf{x}}^{(1)}\alpha_{2, 2} - \overline{\mathbf{x}}^{(2)}\alpha_{2, 1}}{\alpha_{1, 1}\alpha_{2, 2} - \alpha_{1, 2}\alpha_{2, 1}}$
- Thus the source image can be represented as a simple linear combination of the outputs and attacker job reduces to the task of guessing the two weighting coefficients
- Given the strong priors on natural images this seems to be a very tractable problem.

Additional Comments:

- There are three potential ways to work around to address the above comments
  - Devise a new proof for information theoretic security
  - Argue the computational hardness of the search problem
  - Devise a new blinding scheme that does not have this issue

---

> ### Author Response · Authors · 2020-11-14
> **Discussion Regarding the Proof**
>
> We greatly appreciate that you read the proof.
>
> All pixels and Strong Priors: The proof has to be viewed in the context of a single-pixel of an image. Hence, each input $x^i$ is a pixel within the image. For each K inputs (where each input is a pixel of one image) we use a different random noise and hence the encoded output $\bar{x}^i$ can extremely low mutual information (based on the variance of the noise) with respect to the input pixel that it encodes (as we quantified in Appendix A). Given that each pixel in an image is encoded with different random noise, even when pixels have a strong correlation (think two BLUE SKY images being encoded), the mutual information across different pixels can be also significantly eliminated with our encoding approach.
>
> To summarize, for each pixel the information leakage proof holds. We didn’t claim that the total leaked information is bounded by the sum of information leaked by each pixel in an image. Although the pixels are dependent, since we use independent noise with large enough variance, the mutual information between blinded pixels can get arbitrarily small based on the strength of the noise (random scalars, and random noise) used. Note that there is not any method that can make the information leakage exactly zero. Even in the one-time pad as the most reliable method so far, we don’t have zero leakage. This is the first work of its kind that precisely and rigorously characterizes the amount of information leakage under the matrix masking model.
>
> Example: For the specific example you listed,
> In this scenario, the attacker should guess four floating numbers such that the solution$x^{(1)}$ belongs to a highly non-convex complicated set.
> In our work, we have drawn $\alpha_{i,j}$’s from standard Gaussian distribution for simplicity. But in practice, especially when K is small, one can choose $\alpha_{i,j}$’s arbitrarily large to make the task of recovering the image infeasible.
> Computational Hardness: According to the IEE754 standard, an FP32 represents values between $10^{-38}$ to ~$10^{38}$ in $10^{-7}$ increments. So the adversary has to guess  “4” floating-point numbers from this range. Finding these combinations need around $10^{36}$ operations. With the current computation power of fast GPUS (14.90 TFLOPS for Nvidia Titan), it will take hundreds of millions of years to just decode one image correctly.
> In addition to the hardness problem above, the amount of leakage is rigorously bound and the omniscience adversary can’t gain more information than what our proof provides. We hope we convinced you of the hardness and improbability of revealing data with DarKnight.
>
> We appreciate your thoughts and looking forward to more discussions and improving our work.

---

### Official Review · AnonReviewer1 · 2020-11-01
**Review of Reviewer 1**

**Rating:** 4
**Confidence:** 4

**Review:**



Results - The two main proposed benefits of the approach are increased speedups in inference as well as training over  Slalom and just SGX. Looking at Figure 2 and Figure 4, the spped-ups do not appear to be consistently significant. I do appreciate that the authors show the results for MobileNetV2, for which the proposed algorithm is supposed to have less improvement in training time. Overall, I think, given that the main selling point of the time is faster training and inference, it fails to provide a reliable boost in either of them.

Experimental Comparisons - I would have appreciated a comparison with other methods especially those that allow training while maintaining privacy.  While the authors make a note of them in Table 1, their differences with the proposed approach and where one is supposed to be better than the other is discussed neither conceptually nor experimentally (except Slalom)


Novelty - The main algorithmic novelty of the paper is exporting the compute-heavy linear computes to a GPU outside the secure enclave by using blinding-unblinding techniques to protect the privacy.  In my opinion,  while it is a very nice and concise thing to do (along with the results on its privacy guarantee), it is not significantly impactful or interesting to provide enough novelty to this paper. If this would have led to a huge increase in performance margins, I would have said a simple solution that leads to dramatic increase in performance is amazing! However, that is not the case either.

So, given these reasons I am inclined to reject at this point. However, I would encourage the authors to discuss the other techniques in more detail, compare with them and point out situations where this technique can have a larger impact.

=====

I have read the authors' response and my comments remain the same as above especially the paragraph regarding novelty. I am keen to see a revised version of this paper.

---

> ### Author Response · Authors · 2020-11-14
> **Suggestions will be addressed**
>
> Thanks for your comments.
>
> Inference Speedup: We agree that inference speedup over Slalom may be smaller. But the main contribution of DarKnight is to provide “training on private data,” which is not supported in Slalom for the reasons explained in Appendix E.  For training, we achieve 10X speedup for VGG16 over the baseline that trains only on SGX.
>
> Prior Works: To the best of our knowledge, DarKnight is the only work that allows one to use GPUs+SGX to train while preserving data privacy. Since the focus is on training, we can’t rely on finite field arithmetic with quantization since training losses accumulate (which is why training is done using FP).  None of the prior frameworks that use SGX can collaboratively execute with GPU and hence their performance is bound by the CPU performance. We will add a more comprehensive literature review section in the Appendix in our final version.

---

### Author Response · Authors · 2020-11-14
**A few key points to mention**

Dear Reviewers,
We appreciate your feedback in improving our work. A few key points to mention:

1)The primary goal of DarKnight is to “Train” DNNs using private data. We start with how inference can be done with DarKnight so readers can get a quick flavor of our approach before diving deeper into the training process.

2)Training with finite field arithmetic is challenging due to precision losses. Hence we preserved the use of floating-point values in our approach. In this regard, we must reiterate that the mutual information metric is not an average or approximate metric. It has a rigorous basis in information theory. A value of zero, for instance, indicates that the input pixel value and the output pixel value are completely independent.

3)DarKnight guarantees that leakage in bits is less than the roundoff errors in floating-point arithmetic. Perfect privacy at a given precision is when the information leakage is less than the specified precision (precision being limited due to roundoff error) (Guo et al., 2020). In this work, the perfect privacy is achieved at a precision of about 10^-6 in the IEEE standard single-precision arithmetic.

---

### Decision · Program_Chairs · 2021-01-07
**Final Decision**

**Decision:**

Reject

**Comment:**

While reviewers appreciated the simple approach of this work, the biggest concern reviewers had was with the security guarantee of the method. R4 argued that in a certain case recovering an original image x_1 amounted to guessing 2 coefficients. In the discussion phase the authors argued that security amounts to the adversary guessing 4 floating point numbers, not 2, which requires 100s of millions of years to decode an image correctly. However, R4 is correct that only 2 floating point numbers are necessary. This is because, as described by R4 when one sees outputs x_1 * a_{2,2} and x_2 * a_{2,1}, they can reconstruct x_1 as:

x_1 = (x_1 * a_{2,2} - x_2 * a_{2,1}) / (a_{1,1} * a_{2,2} - a_{1,2} * a_{2,1})

Now define:

b_1 := a_{2,2} / (a_{1,1} * a_{2,2} - a_{1,2} * a_{2,1})
b_2 := a_{2,1} / ((a_{1,1} * a_{2,2} - a_{1,2} * a_{2,1})

Thus the above equation can be written as:

x_1 = x_1 * b_1 - x_2*b_2

So an adversary needs to guess 2 floating point numbers. Further, R4 points out that an adversary can obtain x_1 up to a scale factor by simply guessing the relative ratio of the the 2 unknown floating point numbers, i.e., if our guess is c:

x_1/c = x_1 * (b_1/c) - x_2 * (b_2/c)

This is a single floating point number, and not all floating point numbers need to be checked. For many images, information can be leaked even if the true scale of the image is not known.

For this reason I would urge the authors to strengthen the security guarantee of their approach. One way to do this would be to adapt the method so to make the resulting guarantee be a more standard one (e.g., differential privacy, standard cryptographic hardness guarantees). This would eliminate the main reviewer concerns and greatly strengthen the paper.